# PROCEEDINGS A

fluid mechanics, applied mathematics, mathematical physics

Liouville equation, vortex dynamics, Stuart vortex, ring torus, elliptic functions

**Author for correspondence:**
Takashi Sakajo
e-mail: sakajo@math.kyoto-u.ac.jp

# Exact solution to a Liouville equation with Stuart vortex distribution on the surface of a torus

## Takashi Sakajo

Department of Mathematics, Kyoto University, Kitashirakawa Oiwake-cho, Kyoto 606-8502, Japan

TS, 0000-0002-4290-0942

A steady solution of the incompressible Euler equation on a toroidal surface $\mathbb{T}_{R,r}$ of major radius $R$ and minor radius $r$ is provided. Its streamfunction is represented by an exact solution to the modified Liouville equation, $\nabla^2_{\mathbb{T}_{R,r}} \psi = c\, \mathrm{e}^{d\psi} + (8/d)\kappa$, where $\nabla^2_{\mathbb{T}_{R,r}}$ and $\kappa$ denote the Laplace–Beltrami operator and the Gauss curvature of the toroidal surface respectively, and $c, d$ are real parameters with $cd < 0$. This is a generalization of the flows with smooth vorticity distributions owing to Stuart (Stuart 1967 *J. Fluid Mech.* **29**, 417–440. (doi:10.1017/S0022112067000941)) in the plane and Crowdy (Crowdy 2004 *J. Fluid Mech.* **498**, 381–402. (doi:10.1017/S0022112003007043)) on the spherical surface. The flow consists of two point vortices at the innermost and the outermost points of the toroidal surface on the same line of a longitude, and a smooth vorticity distribution centred at their antipodal position. Since the surface of a torus has non-constant curvature and a handle structure that are different geometric features from the plane and the spherical surface, we focus on how these geometric properties of the torus affect the topological flow structures along with the change of the aspect ratio $\alpha = R/r$. A comparison with the Stuart vortex on the flat torus is also made.

## 1. Introduction

For the velocity field $(u(x, y), v(x, y))$ of the incompressible fluid in the two-dimensional plane $\mathbb{R}^2$, the streamfunction $\psi(x, y)$ is introduced as $u = \partial_y \psi$ and $v = -\partial_x \psi$ so

that the incompressibility condition $\partial_x u + \partial_y v = 0$ is satisfied. When the vorticity $\omega(x, y)$ is defined by $\omega = \partial_x v - \partial_y u$, these two functions are connected through the Poisson equation, $\nabla^2 \psi = \partial_x^2 \psi + \partial_y^2 \psi = -\omega$. In the study of fluid dynamics, by assuming the functional form of the vorticity $\omega$ in this equation, one can consider various kinds of flows with vortex structures. For a point measure distribution at $(x_0, y_0) \in \mathbb{R}^2$, i.e. $\omega(x, y) = \delta_{(x_0, y_0)}(x, y)$, its corresponding vortex structure is called a *point vortex*. The constant vorticity distribution on a domain $D \subset \mathbb{R}^2$ gives rise to a *vortex patch*, in which the non-zero vorticity region is enclosed by the boundary $\partial D$ of the domain. More generally, when the vorticity is represented by a smooth function of the streamfunction, say $-\omega = F(\psi)$, the solution to the quasi-linear elliptic equation $\nabla^2 \psi = F(\psi)$ yields the steady solution to the two-dimensional Euler equation. Many steady solutions of the Euler equations with vortex structures are found in the monographs [1,2].

Suppose now that the smooth function is specified by $F(\psi) = c\,e^{d\psi}$ for $c, d \in \mathbb{R}$. We then have the following equation, called the Liouville equation [3],

$$\nabla^2 \psi = c\,e^{d\psi}. \tag{1.1}$$

This equation appears not only in fluid dynamics but also in many problems of mathematical physics such as the field theory and plasma physics. Hence, its exact solutions have been obtained with many mathematical methods. See, e.g., the papers [4–6] for known exact solutions and the references related to this equation. As an example of the applications to fluid dynamics, by regarding the Liouville equation as a mathematical model of free shear layers, Stuart [7] obtained a periodic row of smooth vorticity distributions, which is known as *Stuart vortex*. In connection with geophysical and astrophysical flows, Crowdy [8] extended the notion of Stuart vortex on the surface of the unit sphere $\mathbb{S}^2$. The Liouville equation is then modified as $\nabla_{\mathbb{S}^2}^2 \psi = c\,e^{d\psi} + 2/d$, where $\nabla_{\mathbb{S}^2}^2$ denotes the Laplace–Beltrami operator on the surface. Owing to this modification, he successfully obtained the steady solution of the incompressible Euler equations on the spherical surface consisting of a finite number of smooth vortex structures spaced equally along the line of a latitude and the two point vortices at both poles.

The purpose of the present study is constructing Stuart vortex on the surface of a torus, a compact surface having different geometric features from the plane and the sphere. Since vortex dynamics on the toroidal surface has recently been formulated with using the analytic representation of Green's function on the surface [9], a few steady flows with vortex structures are known up to now: it has been shown in [10] that point vortices located at the antipodal positions, and a polygonal ring configuration of identical $N$ point vortices along the line of a latitude become point vortex equilibria.

More equilibrium states of point vortices whose relative configuration is unchanged throughout the evolution, called *vortex crystals*, are obtained [11]. However, on the other hand, no steady solution with a smooth vorticity distribution, even a vortex patch distribution, has not yet been constructed due to lack of established mathematical treatment of the incompressible Euler equation on the toroidal surface. Hence, it is a mathematical challenge to find such steady flows in order to examine how the geometric properties of the toroidal surface such as non-constant curvature and the existence of a handle structure affect the flow structures. Also, the present study shall provide an insight into the mathematical generalizations of the Liouville equation on general compact Riemannian surfaces. In particular, it is of a mathematical significance to understand how the Liouville equation is to be modified in connection with the conformal structure of the toroidal surface.

In the meantime, naively speaking, there seem to be less relevant to real flow phenomena, since it is difficult to confine fluid flows on the surface of a torus unless we assume the existence of a certain special external forcing. On the other hand, it has been pointed out that the incompressible Euler flow on the toroidal surface is available as a theoretical model of superfluid helium on porous media [12–15]. The readers can find further discussions on the significance of vortex dynamics on surfaces in the survey [16]. Constructing Stuart vortex on the toroidal surface brings us some useful analogies in this physical literature.

The flows on the toroidal surface should be periodic both in the latitudinal as well as the longitudinal directions. In this regard, it is useful to review some preceding results on the steady vortex structures on the doubly periodic flat torus. Stationary lattice structures of point vortices [17–19] and a vortex patch without changing its boundary shape [20] are known. Gurarie & Chow [21] investigated the solution to the sinh-Poisson equation, $\nabla^2 \psi = \sigma \sinh \psi$, in a doubly periodic domain and showed the existence of an elongated 'cat-eye' pattern and a symmetric 'diagonal' configuration with smooth vorticity distributions. While no exact solutions to the Liouville equation on the flat torus have been provided explicitly to the best of our knowledge, we obtain the exact solution to the Liouville equation on the flat torus as a consequence of the present study. Not only that, Stuart vortex on the toroidal surface elucidates a difference from that on the flat torus with respect to the existence of curvature and the Gauss constraint, where the total vorticity over the closed surface vanishes.

The paper is organized as follows. In §2, we give a mathematical formulation of Stuart vortex on the surface of a torus in the same spirit as Crowdy's work for the spherical case [8]. We then derive a modified Liouville equation, in which the relation between the modification term and the geometric conformal structure of the toroidal surface is clarified. In §3, we construct an exact solution to the modified Liouville equation on the toroidal surface. At the same time, the solution to the Liouville equation on the flat torus is obtained. In §4, we investigate the properties of the exact solution with Stuart vortex distribution and compare it with that on the flat torus. Final section gives some discussions for future studies.

## 2. Formulation of Stuart vortex on the surface of a torus

Let $\mathbb{T}_{R,r}$ be the toroidal surface of major radius $R$ and minor radius $r$. The Riemannian metric is introduced through the following three-dimensional Euclidean representation of the surface [9]:

$$\iota : (\theta_m, \phi_m) \in \mathbb{T}_{R,r} \mapsto ((R - r\cos\theta_m)\cos\phi_m, (R - r\cos\theta_m)\sin\phi_m, r\sin\theta_m) \in \mathbb{E}^3. \tag{2.1}$$

The aspect ratio between the two radii is denoted by $\alpha = R/r > 1$, from which we define two real parameters $\mathcal{A} = (\alpha^2 - 1)^{-1/2} > 0$ and $\rho = \exp(-2\pi\mathcal{A}) \in (0, 1)$.

The toroidal surface is endowed with a complex analytic structure through the stereographic projection,

$$\zeta : (\theta, \phi) \in \mathbb{T}_{R,r} \mapsto e^{i\phi} \exp\left(-\int_0^\theta \frac{du}{\alpha - \cos u}\right) \equiv e^{i\phi} \exp(r_c(\theta)), \tag{2.2}$$

in which $r_c(\theta)$ is defined by

$$r_c(\theta) = -\int_0^\theta \frac{du}{\alpha - \cos u} = \log|\zeta|.$$

Note that $r_c(\theta)$ is monotonically decreasing owing to $r_c'(\theta) < 0$ for $\alpha > 1$, and it satisfies a quasi-periodicity in terms of $\theta$ as follows:

$$r_c(\theta \pm 2\pi) = \mp 2\pi\mathcal{A} + r_c(\theta).$$

The Laplace–Beltrami operator $\nabla^2_{\mathbb{T}_{R,r}}$ on the toroidal surface is given by

$$\nabla^2_{\mathbb{T}_{R,r}} \equiv \frac{1}{r^2(R - r\cos\theta)}\frac{\partial}{\partial\theta}\left((R - r\cos\theta)\frac{\partial}{\partial\theta}\right) + \frac{1}{(R - r\cos\theta)^2}\frac{\partial^2}{\partial\phi^2}.$$

According to Green & Marshall [9], we introduce the streamfunction $\psi(\theta, \phi)$ in terms of the toroidal coordinates $(\theta, \phi) \in \mathbb{R}/2\pi\mathbb{Z} \times \mathbb{R}/2\pi\mathbb{Z}$ for the incompressible velocity field $(u_\theta(\theta, \phi), u_\phi(\theta, \phi))$ as follows:

$$u_\theta = \frac{1}{R - r\cos\theta} \frac{\partial\psi}{\partial\phi} \quad \text{and} \quad u_\phi = -\frac{1}{r}\frac{\partial\psi}{\partial\theta}.$$

The vorticity $\omega(\theta, \phi)$ is obtained by taking the curl of the velocity field. Then the streamfunction $\psi$ and the vorticity $\omega$ satisfy the following equation:

$$\nabla^2_{\mathbb{T}_{R,r}} \psi(\theta, \phi) = -\omega(\theta, \phi). \tag{2.3}$$

Furthermore, since $\mathbb{T}_{R,r}$ is a compact surface without boundary, Gauss's divergence theorem gives rise to a global constraint on the vorticity distribution over the toroidal surface, i.e.

$$\iint_{\mathbb{T}_{R,r}} \omega \, d\sigma = 0,$$

in which $d\sigma$ denotes the area element of the toroidal surface. We refer to the constraint as the *Gauss constraint*.

In the present study, we consider a special functional form of the vorticity distribution corresponding to Stuart vortex, namely

$$-\omega(\theta, \phi) = c\, e^{d\psi} + g(\theta, \phi),$$

where $c$, $d$ are arbitrary real parameters with $cd < 0$, and $g(\theta, \phi)$ is a function determined later. Substituting this relation into (2.3), we obtain the following modified Liouville equation on the toroidal surface:

$$\nabla^2_{\mathbb{T}_{R,r}} \psi(\theta, \phi) = c\, e^{d\psi} + g(\theta, \phi). \tag{2.4}$$

In order to construct the exact solution to (2.4) by specifying the function $g(\theta, \phi)$ in an appropriate manner, we review the preceding results on the Liouville equation. Stuart [7] provided one form of general solution $\psi_p(\zeta, \bar{\zeta})$ to the Liouville equation $\nabla^2\psi_p = c\, e^{d\psi_p}$ in the domain $D$ of the complex plane $\zeta \in \mathbb{C}$ as follows:

$$\psi_p(\zeta, \bar{\zeta}) = \frac{1}{d}\log\left[\frac{2f'(\zeta)\bar{f}'(\bar{\zeta})}{-cd(1 + f(\zeta)\bar{f}(\bar{\zeta}))^2}\right], \tag{2.5}$$

where $f(\zeta)$ is an analytic function on $D$. In its extension to the general solution on the spherical surface, Crowdy [8] considered a modified Liouville equation,

$$\nabla^2_{\mathbb{S}^2}\psi_s = c\, e^{d\psi_s} + \frac{2}{d}.$$

Owing to the existence of the additive constant $2/d$, he successfully derived an analytic formula of the exact solution $\psi_s$ to this modified Liouville equation from the exact solution (2.5) to the planar Liouville equation (1.1). In the same spirit as Crowdy's work, we determine the function $g(\theta, \phi)$ so that the exact solution to (2.4) is constructed from the solution to the Liouville equation (1.1).

Let us provide the complex representation of the Laplace–Beltrami operator $\nabla^2_{\mathbb{T}_{R,r}}$ by using the change of variables $(\zeta, \bar{\zeta}) \leftrightarrow (\theta, \phi)$ through the stereographic projection (2.2). Owing to

$$\left.\frac{\partial}{\partial\theta}\right|_\phi = -\frac{\zeta}{\alpha - \cos\theta}\frac{\partial}{\partial\zeta} - \frac{\bar{\zeta}}{\alpha - \cos\theta}\frac{\partial}{\partial\bar{\zeta}} \quad \text{and} \quad \left.\frac{\partial}{\partial\phi}\right|_\theta = i\zeta\frac{\partial}{\partial\zeta} - i\bar{\zeta}\frac{\partial}{\partial\bar{\zeta}},$$

we obtain

$$\nabla^2_{\mathbb{T}_{R,r}} = \frac{4|\zeta|^2}{(R - r\cos\theta)^2}\frac{\partial}{\partial\zeta}\frac{\partial}{\partial\bar{\zeta}}. \tag{2.6}$$

Suppose that we have the exact solution $\psi_p(\zeta, \bar{\zeta})$ to the Liouville equation (1.1) on the annular domain,

$$D_\zeta = \{\zeta \in \mathbb{C} \mid \rho < |\zeta| < 1\}.$$

We then consider the following ansatz for the solution to the modified Liouville equation (2.4):

$$\psi(\zeta, \bar{\zeta}) = \psi_p(\zeta, \bar{\zeta}) - \frac{2}{d} \log\left[\frac{R - r\cos\theta}{2|\zeta|}\right].$$

(2.7)

The second term is introduced so that the conformal factor $4|\zeta|^2/(R - r\cos\theta)^2$ in the Laplace–Baltrami operator (2.6) is eliminated as we see in what follows. From (2.7), we simply have

$$e^{d\psi} = e^{d\psi_p}\left[\frac{4|\zeta|^2}{(R - r\cos\theta)^2}\right]$$

and

$$\partial_{\bar{\zeta}}\partial_{\zeta}\psi = \partial_{\bar{\zeta}}\partial_{\zeta}\psi_p - \partial_{\bar{\zeta}}\partial_{\zeta}\left[\frac{2}{d}\log(R - r\cos\theta) - \frac{2}{d}\log|\zeta|\right]$$

$$= c\,e^{d\psi_p} - \partial_{\bar{\zeta}}\left[-\frac{2}{d}\frac{\alpha - \cos\theta}{\zeta}\frac{r\sin\theta}{R - r\cos\theta} - \frac{1}{d\zeta}\right]$$

$$= c\,e^{d\psi_p} - \partial_{\bar{\zeta}}\left[-\frac{2}{d\zeta}\sin\theta - \frac{1}{d\zeta}\right] = c\,e^{d\psi_p} - \frac{2}{d\zeta}\frac{\alpha - \cos\theta}{\bar{\zeta}}\cos\theta$$

$$= c\,e^{d\psi_p} - \frac{2}{d|\zeta|^2}(\alpha - \cos\theta)\cos\theta,$$

since $\psi_p$ is the solution to the Liouville equation (1.1). Substituting them into (2.4), we have

$$\frac{4|\zeta|^2}{(R - r\cos\theta)^2}\left(c\,e^{d\psi_p} - \frac{2}{d|\zeta|^2}(\alpha - \cos\theta)\cos\theta\right) = c\,e^{d\psi_p}\left[\frac{4|\zeta|^2}{(R - r\cos\theta)^2}\right] + g(\theta, \phi).$$

Hence, the function $g$ becomes a function of $\theta$, specified by

$$g(\theta) = \frac{4|\zeta|^2}{(R - r\cos\theta)^2}\left(-\frac{2}{d|\zeta|^2}(\alpha - \cos\theta)\cos\theta\right) = -\frac{8}{d}\frac{\cos\theta}{r(R - r\cos\theta)}.$$

Hence, the modified Liouville equation on the toroidal surface is given by

$$\nabla^2_{\mathbb{T}_{R,r}}\psi = c\,e^{d\psi} - \frac{8\cos\theta}{dr(R - r\cos\theta)}.$$

(2.8)

Let us remark that the modification term in the Liouville equation on the toroidal surface becomes the function of $\theta$, whereas it is the constant $2/d$ for the spherical case. The discrepancy is solved, since those additive terms are related to the Gauss curvature of the surfaces. As a matter of fact, with the Gauss curvature $\kappa(\mathbb{T}_{R,r})$ of the toroidal surface, the modification function $g(\theta)$ is represented by

$$g(\theta) = \frac{8}{d}\kappa(\mathbb{T}_{R,r}),$$

whereas it is just represented by $(2/d)\kappa(\mathbb{S}^2)$ since the curvature of the surface of the unit sphere is $\kappa(\mathbb{S}^2) = 1$. Furthermore, we should note that the existence of the modified function $g(\theta)$ is inconsequential to the Gauss constraint since Gauss–Bonnet theorem assures the total curvature over the toroidal surface vanishes:

$$\iint_{\mathbb{T}_{R,r}} \kappa(\mathbb{T}_{R,r})\,d\sigma = 2\pi\chi(\mathbb{T}_{R,r}) = 0,$$

where $\chi(M)$ denotes the Euler characteristic of the manifold $M$. Hence, it is sufficient to find the exact solution satisfying

$$\iint_{\mathbb{T}_{R,r}} e^{d\psi}\,d\sigma = 0.$$

# 3. Exact solution to the modified Liouville equation

We are constructing the exact solution (2.5) to the Liouville equation (1.1) on the annular domain $D_\zeta$ by specifying an analytic function $f(\zeta)$. Crowdy [8] has discussed that the analytic function $f(\zeta)$ does not need to be analytic everywhere on $D_\zeta$. That is to say, even if $f(\zeta)$ has isolated pole singularities, the solution (2.5) remains non-singular. On the other hand, when there exists a point satisfying $f'(\zeta) = 0$ in the domain, the solution (2.5) acquires a logarithmic singularity. However, this problem will be resolved in a physically admissible manner as discussed later.

The analytic function $f(\zeta)$ must be chosen so that the streamfunction $\psi(\zeta, \bar{\zeta})$ becomes doubly periodic on $D_\zeta$, since it is a function defined on the toroidal surface. To be specific, $\psi(\zeta, \bar{\zeta})$ needs to be invariant under the transformations $\zeta \mapsto \rho^n \zeta$ and $\phi \mapsto \phi + 2\pi m$ for any $n, m \in \mathbb{Z}$. With the conformal mapping $z = \log \zeta = r_c(\theta) + i\phi$ for $(\theta, \phi) \in \mathbb{R}/2\pi\mathbb{Z} \times \mathbb{R}/2\pi\mathbb{Z}$, it is equivalent to say that $\psi(z, \bar{z})$ is a doubly periodic function on the domain $\{z \in \mathbb{C} \mid -2\pi\mathcal{A} \leqq \operatorname{Re} z \leqq 0,\ 0 \leqq \operatorname{Im} z \leqq 2\pi\}$ with the aspect ratio $1/\mathcal{A} = \sqrt{\alpha^2 - 1}$. In order to confirm the double periodicity of the streamfunction $\psi(z, \bar{z})$, we can use

$$D_z = \{z \in \mathbb{C} \mid 0 \leqq \operatorname{Re} z \leqq 2\pi\mathcal{A},\ 0 \leqq \operatorname{Im} z \leqq 2\pi\}$$

as its fundamental domain without loss of generality. We also note that the domain $D_z$ is identified with the two-dimensional domain $(\theta, \phi) \in \mathbb{R}/2\pi\mathbb{Z} \times \mathbb{R}/2\pi\mathbb{Z}$, since $r_c(\theta): [0, 2\pi] \to [-2\pi\mathcal{A}, 0]$ is one-to-one owing to $r_c'(\theta) < 0$ for $\alpha > 1$. Then the streamfunction $\psi(\theta, \phi)$ becomes invariant with respect to the transformations $\theta \mapsto \theta + 2\pi n$ and $\phi \mapsto \phi + 2\pi m$ for $n, m \in \mathbb{Z}$.

Let us remember that Stuart [7] used the function $f(z) = \tan(z) = \sin(z)/\cos(z)$ to construct the solution (2.5) with the periodic boundary condition in the real direction. Hence, a natural extension of this function to the doubly periodic case is given by

$$f(z) = \frac{\operatorname{sn}(z)}{\operatorname{cn}(z)} = \frac{\operatorname{sn}(\log \zeta)}{\operatorname{cn}(\log \zeta)},$$

where $\operatorname{sn}(z)$ and $\operatorname{cn}(z)$ are the Jacobi elliptic functions with the quarter periods $K = \pi\mathcal{A}$ and $K' = \pi$. The definitions of those functions and their formulae used in the present paper are found in the handbook [22]. Since the derivatives of $\operatorname{sn}(\log \zeta)$ and $\operatorname{cn}(\log \zeta)$ are given by

$$\frac{\mathrm{d}}{\mathrm{d}\zeta}\operatorname{sn}(\log \zeta) = \frac{1}{\zeta}\operatorname{cn}(\log \zeta)\operatorname{dn}(\log \zeta) \quad \text{and} \quad \frac{\mathrm{d}}{\mathrm{d}\zeta}\operatorname{cn}(\log \zeta) = -\frac{1}{\zeta}\operatorname{sn}(\log \zeta)\operatorname{dn}(\log \zeta),$$

where $\operatorname{dn}(z)$ denotes the other Jacobi elliptic function, we have

$$\frac{\mathrm{d}}{\mathrm{d}\zeta}f'(\log \zeta) = \frac{1}{\zeta}\frac{\operatorname{dn}(\log \zeta)}{\operatorname{cn}^2(\log \zeta)}.$$

Consequently, the exact solution (2.5) on the domain $D_\zeta$ to the Liouville equation (1.1) becomes

$$\psi_p(\zeta, \bar{\zeta}) = \frac{1}{d}\log\left[-\frac{2|\operatorname{dn}(\log \zeta)|^2}{cd(|\operatorname{sn}(\log \zeta)|^2 + |\operatorname{cn}(\log \zeta)|^2)^2|\zeta|^2}\right],$$

from which we obtain the solution (2.7) to the modified Liouville equation (2.8) on the toroidal surface.

$$\psi(\zeta, \bar{\zeta}) = \frac{1}{d}\log\left[-\frac{8|\operatorname{dn}(\log \zeta)|^2}{cd(|\operatorname{sn}(\log \zeta)|^2 + |\operatorname{cn}(\log \zeta)|^2)^2(R - r\cos\theta)^2}\right]. \tag{3.1}$$

In terms of $z = \log \zeta = r_c(\theta) + i\phi$, it is represented by

$$\psi(z, \bar{z}) = \frac{1}{d}\log\left[-\frac{8|\operatorname{dn}(z)|^2}{cd(|\operatorname{sn}(z)|^2 + |\operatorname{cn}(z)|^2)^2(R - r\cos\theta)^2}\right]. \tag{3.2}$$

We confirm that $\psi(z,\bar{z})$ is a doubly periodic function on the fundamental domain $D_z$. Since $R - r\cos\theta$ is periodic with respect to $\theta \mapsto \theta - 2\pi$, i.e. $z \mapsto z + 2\pi\mathcal{A} = z + 2K$ owing to the quasi-periodicity of $r_c(\theta)$, it is sufficient to see that the function

$$w(z,\bar{z}) = \frac{|\mathrm{dn}(z)|^2}{(|\mathrm{sn}(z)|^2 + |\mathrm{cn}(z)|^2)^2}$$

is invariant with respect to the transformations $z \mapsto z + 2K$ and $z \mapsto z + 2iK'$ with $K = \pi\mathcal{A}$ and $K' = \pi$. As a matter of fact, it follows from

$$\mathrm{sn}(z + 2K) = -\mathrm{sn}(z), \quad \mathrm{cn}(z + 2K) = -\mathrm{cn}(z) \quad \text{and} \quad \mathrm{dn}(z + 2K) = \mathrm{dn}(z),$$

and

$$\mathrm{sn}(z + 2iK') = \mathrm{sn}(z), \quad \mathrm{cn}(z + 2iK') = -\mathrm{cn}(z) \quad \text{and} \quad \mathrm{dn}(z + 2iK') = -\mathrm{dn}(z),$$

that $w(z,\bar{z})$ becomes doubly periodic with periods $2K$ and $2iK'$ on the fundamental domain $D_z$. It is now easy to check that the streamfunction $\psi(\zeta,\bar{\zeta})$ is doubly periodic on the annular domain $D_\zeta$, since

$$\mathrm{sn}(\log\rho\zeta) = \mathrm{sn}(\log\zeta - 2\pi\mathcal{A}) = \mathrm{sn}(\log\zeta - 2K) = -\mathrm{sn}(\log\zeta),$$

$$\mathrm{cn}(\log\rho\zeta) = \mathrm{cn}(\log\zeta - 2\pi\mathcal{A}) = \mathrm{cn}(\log\zeta - 2K) = -\mathrm{cn}(\log\zeta),$$

$$\mathrm{dn}(\log\rho\zeta) = \mathrm{dn}(\log\zeta - 2\pi\mathcal{A}) = \mathrm{dn}(\log\zeta - 2K) = \mathrm{dn}(\log\zeta)$$

and

$$\mathrm{sn}(\log\zeta(\theta,\phi+2\pi)) = \mathrm{sn}(\log\zeta(\theta,\phi)+2iK') = \mathrm{sn}(\log\zeta),$$

$$\mathrm{cn}(\log\zeta(\theta,\phi+2\pi)) = \mathrm{cn}(\log\zeta(\theta,\phi)+2iK') = -\mathrm{cn}(\log\zeta),$$

$$\mathrm{dn}(\log\zeta(\theta,\phi+2\pi)) = \mathrm{dn}(\log\zeta(\theta,\phi)+2iK') = -\mathrm{dn}(\log\zeta),$$

owing to $\log\zeta(\theta,\phi+2\pi) = \log\zeta(\theta,\phi) + 2\pi\mathrm{i} = \log\zeta(\theta,\phi) + 2iK'$.

In the course of the construction stated above, we have already obtained the solution to the Liouville equation (1.1) on the flat torus. For any fundamental domain with the aspect ratio $1/\mathcal{A} = \sqrt{\alpha^2 - 1}$, plugging $f(z) = \mathrm{sn}(z)/\mathrm{cn}(z)$ into the general formula (2.5) yields the exact solution $\psi_f(z,\bar{z})$ on the domain $D_z$.

$$\psi_f(z,\bar{z}) = \frac{1}{d}\log\left[-\frac{2}{cd}w(z,\bar{z})\right]. \tag{3.3}$$

Comparing (3.3) with (3.1), we clearly see the effect of the curvature as follows:

$$\psi(z,\bar{z}) - \psi_f(z,\bar{z}) = \frac{1}{d}\log\left[\frac{4}{(R - r\cos\theta)^2}\right].$$

We here examine the zeros and the poles of the function $w(z,\bar{z})$. Figure 1 shows the locations of the poles and the zeros of the Jacobi elliptic functions $\mathrm{sn}(z)$, $\mathrm{cn}(z)$ and $\mathrm{dn}(z)$ in the quarter of the fundamental domain $D_z$. We also remember that the order of the zeros and poles are all one. Hence, we find the function $w(z,\bar{z})$ has simple zeros at $\alpha_1 = iK' = \pi\mathrm{i} \sim 2K + iK' = 2\pi\mathcal{A} + \pi\mathrm{i}$ and $\alpha_2 = K + iK' = \pi\mathcal{A} + \pi\mathrm{i}$ in the fundamental domain $D_z$. Accordingly, the streamfunction $\psi(z,\bar{z})$ acquires logarithmic singularities at $z = \alpha_1$ and $z = \alpha_2$. However, this situation is not crucial, since they correspond to point vortices in the physical space. That is to say, according to [9], the streamfunction $\psi_{\mathrm{PV}}(z,\bar{z})$ representing a point vortex located at $z = \alpha_i$ with the strength (circulation) $\Gamma_i$ on the toroidal surface has the following asymptotic expansion:

$$\psi_{\mathrm{PV}}(z,\bar{z}) \sim -\frac{\Gamma_i}{2\pi}\log|z - \alpha_i| + (\text{regular}), \quad z \to \alpha_i.$$

On the other hand, the streamfunction (3.2) has the following expansion in the neighbourhood of $z = \alpha_i$, since $w(z,\bar{z})$ has simple zeros at these points:

$$\psi(z,\bar{z}) \sim \frac{2}{d}\log|z - \alpha_i| + (\text{regular}), \quad z \to \alpha_i.$$

$$iK' \quad \infty \overline{\quad\quad} * \qquad iK' \quad \infty \overline{\quad\quad} * \qquad iK' \quad \infty \overline{\quad\quad} 0$$

$$\left| \begin{array}{c} \\ sn(z) \\ \\ \end{array} \right| * \qquad \left| \begin{array}{c} \\ cn(z) \\ \\ \end{array} \right| * \qquad \left| \begin{array}{c} \\ dn(z) \\ \\ \end{array} \right|$$

$$0 \; 0 \overline{\quad\quad} * \qquad 0 \; * \overline{\quad\quad} 0 \qquad 0 \; * \overline{\quad\quad} *$$
$$\phantom{0\;}0 \quad\quad K \qquad \phantom{0\;}0 \quad\quad K \qquad \phantom{0\;}0 \quad\quad K$$

**Figure 1.** Zeros and poles of the Jacobi elliptic functions sn(z), cn(z) and dn(z) in the quarter of the fundamental domain $D_z$. For each function, the locations of the pole and the zero are represented by the symbols '$\infty$' and '0' respectively. The symbol '$*$' indicates that the function has a non-zero regular value at the point.

Comparing these two leading terms, we have the circulations generated by the point vortices at $z = \alpha_1$ and $z = \alpha_2$ as follows:

$$\Gamma_1 = \Gamma_2 = -\frac{4\pi}{d}.$$

The locations of the two point vortices $z = \alpha_1$ and $\alpha_2$ correspond to the innermost and the outermost points of the toroidal surface on the same line of a longitude $\phi = \pi$ around the handle. Moreover, they are an equilibrium configuration of point vortices as shown in [10]. Hence, $\psi(z, \bar{z})$ gives rise to a physically admissible steady solution to the incompressible Euler equations on the toroidal surface.

We finally confirm whether the solution $\psi(z, \bar{z})$ satisfies the Gauss constraint. The total vorticity over the toroidal surface consists of the circulations generated by the two point vortices at $z = \alpha_1$ and $\alpha_2$, and the contribution from the smooth vorticity distribution. Owing to $\Gamma_1 = \Gamma_2 = -4\pi/d$, the total circulation induced by the two point vortices becomes $-8\pi/d$. The remaining part coming from the smooth vorticity distribution in $w(z, \bar{z})$ is computed analytically in what follows. The area element $d\sigma$ associated with the Euclidean representation (2.1) is given by

$$d\sigma = (R - r\cos\theta)r(\theta)r_c'(\theta)\, d\theta\, d\phi = -(R - r\cos\theta)^2\, d\theta\, d\phi.$$

We here introduce the function $q(z, \bar{z})$ on $D_z$ as

$$q(z, \bar{z}) = \frac{dc(z)\overline{sn(z)}}{|sn(z)|^2 + |cn(z)|^2},$$

where $dc(z) = dn(z)/cn(z)$. This function has simple poles at $z = K = \pi\mathcal{A} \sim K + 2iK' = \pi\mathcal{A} + 2\pi i$ on the boundary of $D_z$ due to the double periodicity. Let us confirm that the residue of $q(z, \bar{z})$ at the poles is $-1$. The Jacobi elliptic functions have the following expansions in the neighbourhood of $u = 0$:

$$sn(u) = u + O(u^3), \quad cn(u) = 1 + O(u^2) \quad \text{and} \quad dn(u) = 1 + O(u^2).$$

For one of the quarter periods $K$, they also satisfy

$$sn(u + K) = \frac{cn(u)}{dn(u)}, \quad cn(u + K) = -k'\frac{sn(u)}{dn(u)} \quad \text{and} \quad dn(u + K) = k'\frac{1}{dn(u)}.$$

Here, $k'$ is the complementary modulus of the Jacobi elliptic functions satisfying $K' = K(k')$, where $K(k)$ denotes the complete elliptic integral of the first kind. Using these formulae, we have the expansion of $q(u + K) = -1/u + O(u)$. Setting $z = u + K$, we have

$$q(z, \bar{z}) = -\frac{1}{z - K} + O(z - K),$$

which yields the result. Let us also note that

$$\partial_{\bar{z}}q(z, \bar{z}) = w(z, \bar{z}).$$

Hence, it follows from Green's formula and the double periodicity of $q(z, \bar{z})$ that we compute the contribution from the smooth vorticity distribution.

$$
\begin{aligned}
-\iint_{\mathbb{T}_{R,r}} c\, e^{d\psi}\, d\sigma &= -\iint_{D_z} \frac{8}{d} w(z)\, d\theta\, d\phi \\
&= -\frac{8}{d} \iint_{D_z} \partial_{\bar{z}} \left( \frac{dc(z)\overline{sn(z)}}{|sn(z)|^2 + |cn(z)|^2} \right) d\theta\, d\phi \\
&= -\frac{4}{id} \oint_{\partial D_z} \frac{dc(z)\overline{sn(z)}}{|sn(z)|^2 + |cn(z)|^2}\, d\theta\, d\phi \\
&= -\frac{4}{id} 2\pi i \operatorname*{Res}_{z=\pi\mathcal{A}} q(z, \bar{z}) = \frac{8\pi}{d}.
\end{aligned}
$$

In total, the vorticity over the toroidal surface satisfies the Gauss constraint.

# 4. Stuart vortex on the toroidal surface

In order to examine the flow generated by the exact solution (3.1), we visualize the streamfunction $\psi(\theta, \phi)$ and the vorticity $\omega(\theta, \phi)$ in the fundamental domain $(\theta, \phi) \in \mathbb{R}/2\pi\mathbb{Z} \times \mathbb{R}/2\pi\mathbb{Z}$. The two real parameters in the modified Liouville equation are set $c = 1$ and $d = -2$. We change the aspect ratio $\alpha = R/r$. The Jacobi elliptic functions $sn(z)$, $cn(z)$ and $dn(z)$ are computed numerically with using a software package for Matlab provided by Moiseev [23]. Since the periods of the fundamental domain $D_z$ are $2K = 2\pi\mathcal{A}$ and $2iK' = 2\pi i$, we set $\tau = iK'/K = i/\mathcal{A} = i\sqrt{\alpha^2 - 1}$. Then the modulus $k$ of the Jacobi elliptic functions is determined by

$$
k = \frac{\theta_2^2(0; \tau)}{\theta_3^2(0; \tau)},
$$

where $\theta_2(v; \tau)$ and $\theta_3(v; \tau)$ are the Jacobi theta functions of modulus $\tau$. We can approximate the modulus $k$ accurately by using the following convergent series with respect to $h = e^{i\pi\tau} = e^{-\pi\sqrt{\alpha^2 - 1}}$:

$$
\theta_2(0; \tau) = 2h^{1/4} + 2h^{\frac{9}{4}} + 2h^{25/4} + \cdots \quad \text{and} \quad \theta_3(0; \tau) = 1 + 2h + 2h^4 + 2h^9 + \cdots.
$$

Figure 2 shows the contour plots of the streamfunction $\psi(\theta, \phi)$, which are streamlines, in the fundamental domain for $\alpha = 1.2$, 1.5, 2.0 and 3.0. For all panels, there are two point vortices at $(0, \pi) \sim (2\pi, \pi)$ and $(\pi, \pi)$ that correspond to the innermost and the outermost points of the toroidal surface on the longitudinal line $\phi = \pi$. Their circulations $\Gamma_1 = \Gamma_2 = -4\pi/d = 2\pi$ are equivalent and independent on $\alpha$, generating counter-clockwise rotational flows around the point vortices. We observe another rotational flow domain centred at the corner point $(\theta, \phi) = (0, 0) \sim (0, 2\pi) \sim (2\pi, 0) \sim (2\pi, 2\pi)$ of the fundamental domain. The corner point is antipodal to the positions of the two point vortices. There exist three saddle points: two of them are on the longitudinal line $\phi = \pi$ between the two point vortices and the other one is located at $(\theta, \phi) = (\pi, 0) \sim (\pi, 2\pi)$ on the boundary of the fundamental domain. The streamlines except the point vortices, the corner point, the saddle points and their connecting orbits represent periodic orbits. Topologically, these periodic orbits are characterized into three categories in terms of the generators, say $\sigma$ and $\mu$, of the fundamental group associated with the toroidal surface.

— *null homotopic periodic orbits*, which are homotopic to a point;
— *$\sigma$-homotopic periodic orbits*, which are homotopic to a loop going around the handle of the torus in the $\theta$-direction;
— *$\mu$-homotopic periodic orbits*, which are homotopic to a loop going around the hole of the torus in the $\phi$-direction.

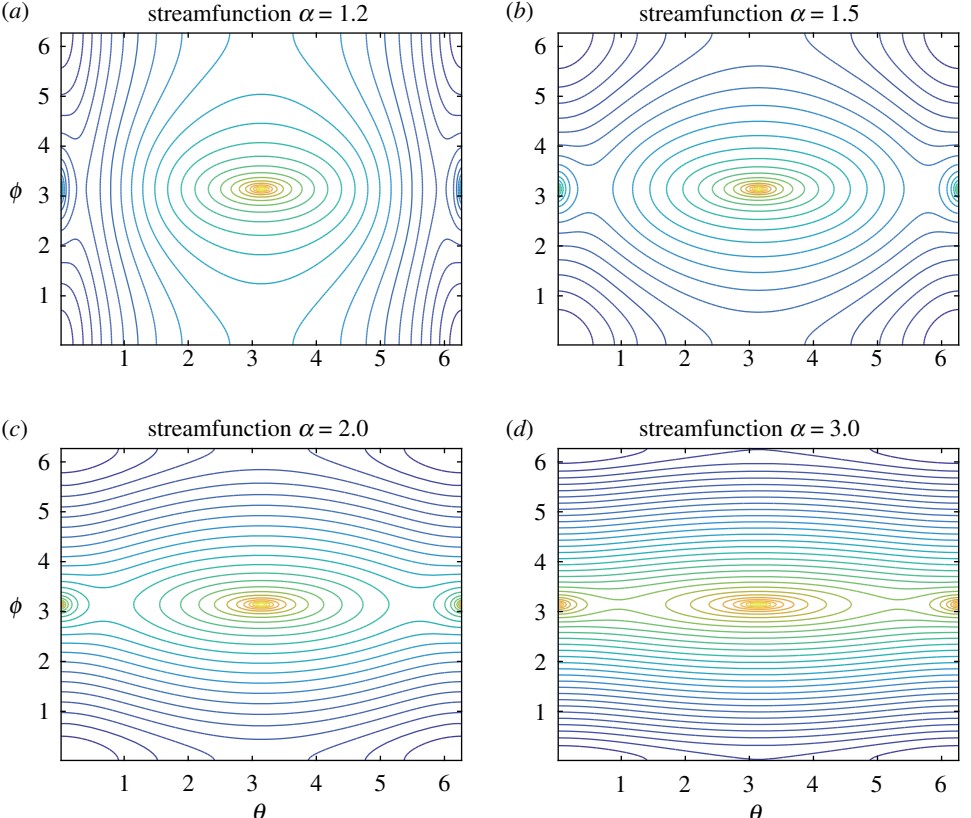

**Figure 2.** Streamlines of the streamfunction $\psi(\theta, \phi)$ in the fundamental domain $(\theta, \phi) \in \mathbb{R}/2\pi\mathbb{Z} \times \mathbb{R}/2\pi\mathbb{Z}$ for the aspect ratios (a) $\alpha = 1.2$, (b) $\alpha = 1.5$, (c) $\alpha = 2.0$, (d) $\alpha = 3.0$. They are the solutions to the modified Liouville equation (2.8) with $c = 1$ and $d = -2$. (Online version in colour.)

The topological structure of separatrices connecting these saddle points changes as $\alpha$ varies. When $\alpha = 1.2$, the saddle points on the line $\phi = \pi$ are connected by a homoclinic and a heteroclinic orbits, whereas the saddle point on the boundary is self-connected. These connecting orbits divide the fundamental domain into small domains containing periodic orbits. There are three domains consisting of null homotopic periodic orbits around the point vortices and the corner point. We observe the other two domains filled with $\mu$-homotopic periodic orbits between the separatrices. At $\alpha = 1.5$, the reconnection of these separatrices occurs. As a result, the two saddle points on the longitudinal line $\phi = \pi$ are now connected by heteroclinic orbits, while the other saddle point at the boundary remains self-connected but its separatrices are running in the $\theta$-direction. Consequently, there appear two domains consisting of $\sigma$ homotopic periodic orbits between the homoclinic and the heteroclinic connections. The streamline topology remains the same qualitatively for $\alpha$ larger than 1.5. However, as the aspect ratio $\alpha$ increases, the three domains of null homotopic periodic orbits deform into thin band-like structures in the $\theta$-direction and the two domains of $\sigma$ homotopic periodic orbits fill the most part of the toroidal surface. See the contour plots of the streamfunction on the surface of the torus in figure 4 to confirm the change of the streamline topology.

In figure 3, we plot the contours of the vorticity,

$$\omega(\theta, \phi) = -c\, e^{d\psi(\theta,\phi)} - \frac{8}{d}\kappa(\mathbb{T}_{R,r}),$$

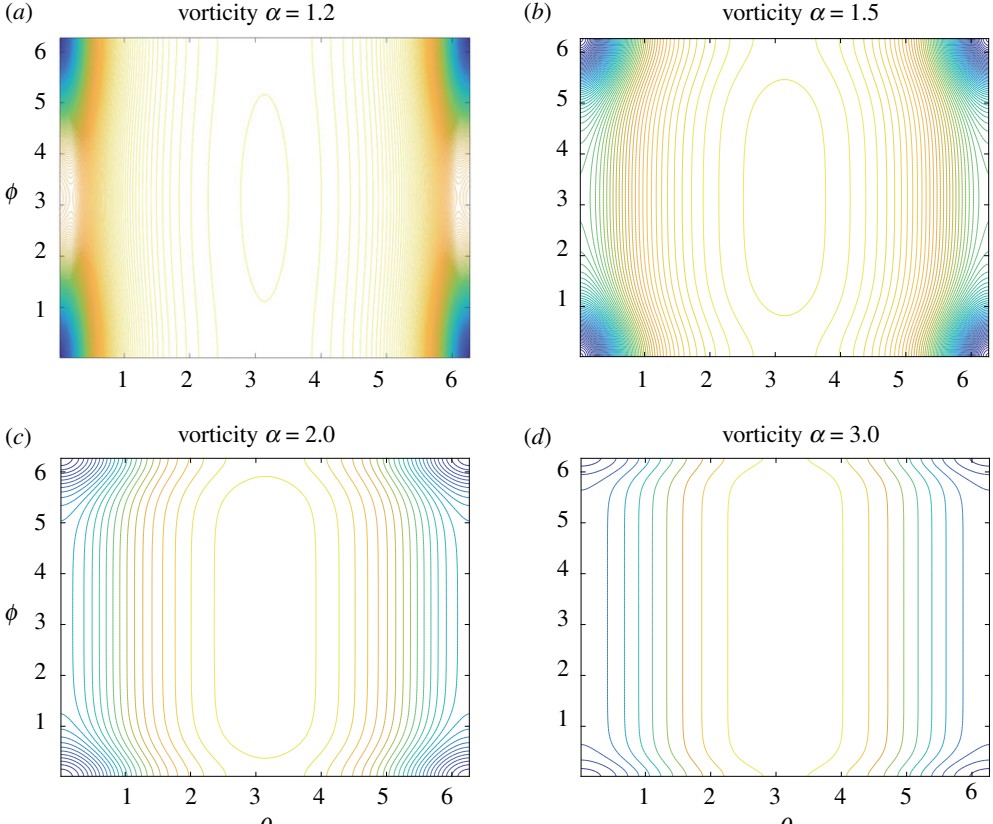

**Figure 3.** Contour plots of the vorticity $\omega(\theta, \phi)$ in the fundamental domain $(\theta, \phi) \in \mathbb{R}/2\pi\mathbb{Z} \times \mathbb{R}/2\pi\mathbb{Z}$ corresponding to the streamfunctions shown in figure 2 for the aspect ratios (a) $\alpha = 1.2$, (b) $\alpha = 1.5$, (c) $\alpha = 2.0$ and (d) $\alpha = 3.0$. (Online version in colour.)

in the fundamental domain $(\theta, \phi) \in \mathbb{R}/2\pi\mathbb{Z} \times \mathbb{R}/2\pi\mathbb{Z}$ for the same parameters as in figure 2. The first term of $\omega(\theta, \phi)$ is the contribution from the streamfunction and the second term represents the effect of curvature of the toroidal surface $\mathbb{T}_{R,r}$. The two point vortices at $(\theta, \phi) = (0, \pi) \sim (2\pi, \pi)$ and $(\pi, \pi)$ generate the positive circulations $2\pi$, but no vorticity is created by them in the whole domain except at their locations where the vorticity diverges. On the other hand, the negative continuous vorticity distribution is observed at the corner of the fundamental domain. When $\alpha$ is small, the negative vorticity at the corner is strongly localized at the innermost part of the toroidal surface. On the contrary, as $\alpha$ gets larger, the localized vorticity distribution at this region gets weaker. Accordingly, the vorticity distribution in the middle of the fundamental domain tends to be dominated by the second term $\omega(\theta, \phi) \sim -(8/d)\kappa(\mathbb{T}_{R,r})$, which depends on $\theta$ in the most part of the toroidal surface, as $\alpha \to \infty$. This behaviour is clearly observed in its contour plots of the vorticity on the surface of the torus in figure 4.

We now observe the Stuart vortex on the flat torus to understand the effect of curvature on the vortex structures. The parameters are the same as $c = 1$ and $d = -2$. In order to adjust the modulus of the fundamental domain for comparison, identifying the complex number $z = x + iy$ with a point $(x, y) \in \mathbb{R}^2$, we determine the range of $x$ and $y$ from a given aspect ratio $0 < \alpha < 1$ as follows. Setting $\tau = i\sqrt{\alpha^2 - 1}$, we have the elliptic modulus $k$ and its complement $k'$ of the Jacobi elliptic functions,

$$k = \frac{\theta_2^2(0; \tau)}{\theta_3^2(0; \tau)} \quad \text{and} \quad k' = \sqrt{1 - k^2}.$$

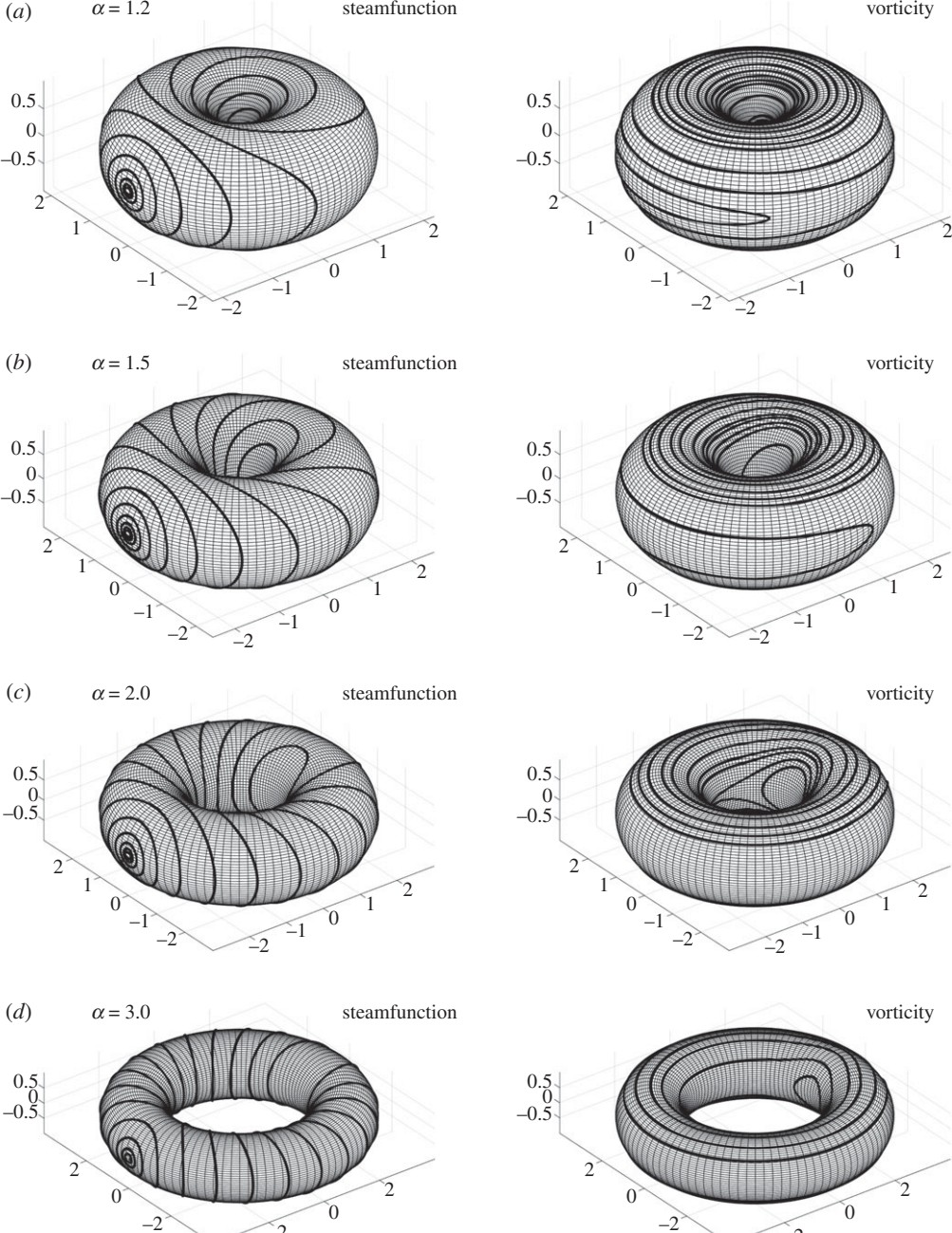

**Figure 4.** Contour plots of the streamfuntion $\psi(\theta, \phi)$ and the vorticity $\omega(\theta, \phi)$ on the surface of the torus with the aspect ratios (a) $\alpha = 1.2$, (b) $\alpha = 1.5$, (c) $\alpha = 2.0$ and (d) $\alpha = 3.0$ corresponding to figures 2 and 3.

Then the fundamental domain with the aspect ratio $\alpha = K(k')/K(k)$ is given by

$$D_z^f = \{(x, y) \in \mathbb{R}^2 \,|\, 0 \leqq x \leqq 2K(k),\ 0 \leqq y \leqq 2K(k')\}.$$

Figure 5 shows the plots of streamlines for the streamfunction $\psi_f(x, y)$ on the flat torus in the fundamental domain $D_z^f$ with the aspect ratios $\alpha = 1.1, 1.3, 1.5$ and $1.9$. As we observe in figure 2, there are two identical point vortices with the circulation $2\pi$ at $(x, y) = (0, K(k')) \sim (2K(k), K(k'))$ and

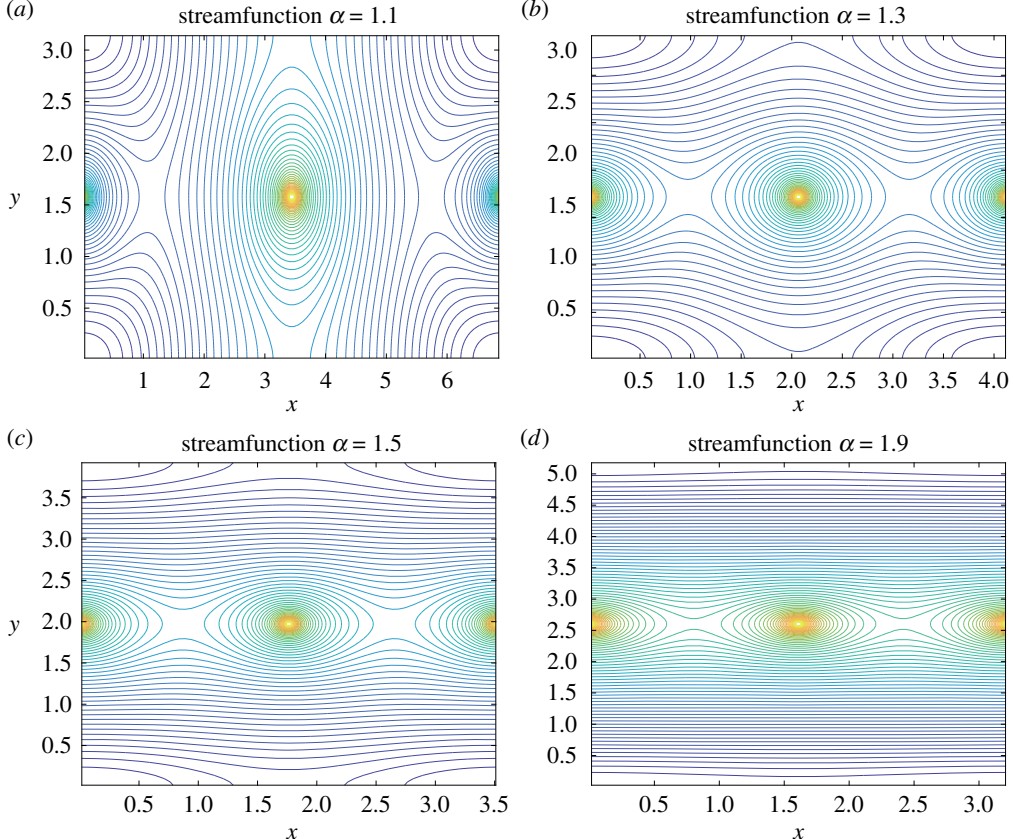

**Figure 5.** Streamlines of the streamfunction $\psi_f(x, y)$ on the flat torus. They are the solutions to the Liouville equation (1.1) in the fundamental domain $D_z^f = [0, 2K(k)] \times [0, 2K(k')]$ with the aspect ratio $\alpha = K(k')/K(k)$ for (a) $\alpha = 1.1$, (b) $\alpha = 1.3$, (c) $\alpha = 1.5$, (d) $\alpha = 1.9$. Here, $K(k)$ denotes the complete elliptic integral of the first kind, and $k$ and $k'$ are the elliptic modulus and its complement, respectively. (Online version in colour.)

$(K(k), K(k'))$ and two saddle points between them on $y = K(k')$. We also observe a saddle point on the boundary $(x, y) = (K(k), 0) \sim (K(k), 2K(k'))$ and a domain consisting of null homotopic periodic orbits centred at the corner of the fundamental domain. As $\alpha$ varies, the topological structure of connecting orbits between three saddle points changes in the same manner as we observed in figure 2. Hence, we see no qualitative difference in the streamline patterns between the cases on the flat torus and the toroidal surface. On the other hand, the change of the streamline patterns for the flat torus occurs at a smaller aspect ratio compared to that for the toroidal surface. Moreover, when $\alpha \gtrsim 1.3$, the flow domains of null homotopic periodic orbits around the two point vortices is homogeneous. This is in contrast to the streamline patterns on the toroidal surface, where the domain of null homotopic periodic orbits around the point vortex at $(\theta, \phi) = (\pi, \pi)$ is larger than that around the point vortex at $(\theta, \phi) = (0, \pi)$. Accordingly, we could say that the inhomogeneity of the domain sizes is due to the effect of the curvature.

Figure 6 is the contour plots of the vorticity $\omega(x, y) = -c\,e^{d\psi_f(x,y)}$ in the fundamental domain $D_z^f$ derived from the streamfunctions in figure 5. Since the two point vortices generate no vorticity around them as noted above, we observe no strong vorticity in the middle of the fundamental domain. The negative vorticity region at the corner is localized for smaller aspect ratios, while it expands in the $x$-direction and forms a thin band-like distribution as the aspect ratio increases. Hence, the vorticity distribution tends to be uniform in the $y$-direction, whereas it still contains a thin vortex core in the centre of the band structure as $\alpha \to \infty$. The asymptotic vorticity distribution

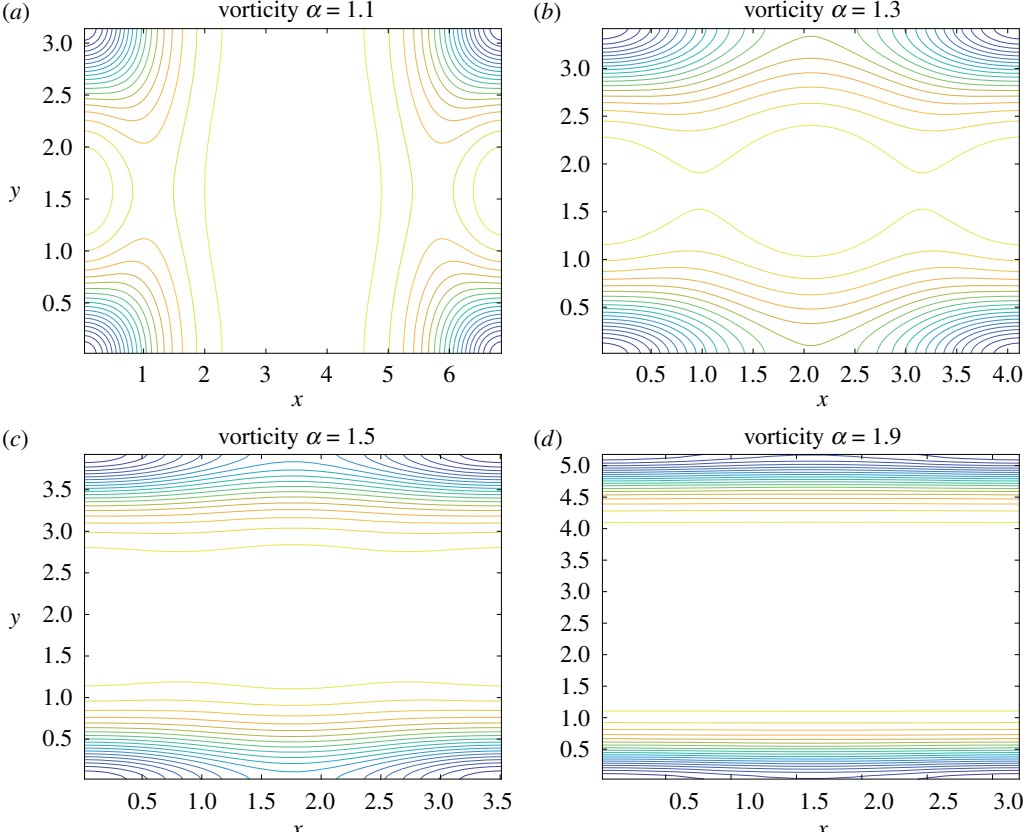

**Figure 6.** Contour plots of the vorticity $\omega(x,y) = -c\,e^{d\psi_f(x,y)}$ corresponding to the streamfunction in figure 5 in the fundamental domain $D_z^f = [0, 2K(k)] \times [0, 2K(k')]$ with the aspect ratio $\alpha = K(k')/K(k)$ for (a) $\alpha = 1.1$, (b) $\alpha = 1.3$, (c) $\alpha = 1.5$, (d) $\alpha = 1.9$. (Online version in colour.)

is different from what we have observed in the case of the Stuart vortex on the toroidal surface, where the vorticity distribution depends strongly on the curvature in the limit. This is another influence by the curvature.

## 5. Discussions

We obtain the analytic formula of an exact solution to the modified Liouville equation,

$$\nabla^2_{\mathbb{T}_{R,r}} \psi = c\,e^{d\psi} + \frac{8}{d}\kappa(\mathbb{T}_{R,r}).$$

This gives rise to a new steady solution of the incompressible Euler equations with the vorticity distribution of Stuart vortex on the toroidal surface $\mathbb{T}_{R,r}$. The solution consists of two identical point vortices located at the innermost and the outermost points of the toroidal surface on the same line of a longitude, and a smooth vortex distribution centred at their antipodal location. The smooth vorticity distribution compensates the circulations generated by the two point vortices so that the Gauss constraint is satisfied. Owing to the non-constant curvature, the domain of null homotopic periodic orbits around the outermost point vortex is larger than that around the innermost one. When the aspect ratio is small, the smooth vorticity distribution is strongly localized. As the aspect ratio gets larger, the corner vorticity distribution weakens and the vorticity distribution thus tends to the Gauss curvature of the toroidal surface. The periodic orbits

are either null homotopic or $\mu$-homotopic for small $\alpha$, but they change to null homotopic and $\sigma$-homotopic periodic orbits for large $\alpha$. The qualitative change of the streamline patterns occurs due to the reconnection of separatrices of the saddle points.

In the generalization of Stuart vortex to the spherical surface owing to Crowdy [8], the constant term $2/d$ needs to be added to the Liouville equation to obtain the exact solution. He argued that it is not clear why the choice of the constant term is relevant, which remains an open problem. The present study strongly suggests that the modification term stems from the Gauss curvature of the surface. That is to say, since the curvature of the spherical surface with the unit radius is $\kappa(\mathbb{S}^2) = 1$, the constant term is rewritten as $(2/d)\kappa(\mathbb{S}^2)$. On the other hand, the modification term for the toroidal surface is given by $(8/d)\kappa(\mathbb{T}_{R,r}) = 4 \cdot (2/d)\kappa(\mathbb{T}_{R,r})$, in which the factor 4 is coming from the definition of the Laplace–Beltrami operator. Hence, it is conjectured that $(2/d)\kappa(\mathcal{M})$ would be the natural choice of the modification term of the Liouville equation for a given compact Riemannian surface $\mathcal{M}$, where $\kappa(\mathcal{M})$ denotes the Gauss curvature of the surface. In addition, according to the Gauss–Bonnet theorem, the Gauss constraint is then rewritten as

$$\int_{\mathcal{M}} c\, e^{d\psi}\, d\sigma + \frac{a}{d} \int_{\mathcal{M}} \kappa(\mathcal{M})\, d\sigma = \int_{\mathcal{M}} c\, e^{d\psi}\, d\sigma + \frac{2\pi a}{d}(2 - 2\mathcal{G}(\mathcal{M})) = 0,$$

where $\mathcal{G}(\mathcal{M})$ represents the genus of the manifold $\mathcal{M}$. It is an interesting mathematical problem to verify this conjecture in future.

In the present paper, we have obtained the exact solution to the modified Liouville equation (2.8) with Stuart vortex distribution having two point vortices. It is more interesting to construct exact solutions having general $N$ point vortices by specifying the other analytic functions $f(\zeta)$. Then, these analytic functions should be doubly periodic in the annular fundamental domain $D_\zeta$. It should also be verified whether the solution satisfies the Gauss condition for every choice of $f(\zeta)$. In addition, if there exist points satisfying $f'(\zeta) = 0$ that correspond to point vortex singularities, we need to confirm if they are an equilibrium solution to the evolution equation of point vortices given in [10]. We note that it is not an easy task to obtain point vortex equilibria on the toroidal surface in general, as shown in [11]. One of the possible candidates is an exact solution with Stuart vortex distribution having the ring configuration of $N$ point vortices along the line of a latitude whose linear stability has been investigated in detail [24]. This will be a future work.

Data accessibility. This article does not contain any additional data.

Authors' contributions. All works are done by T.S.

Competing interests. I declare I have no competing interests.

Funding. This work is supported by JSPS Kakenhi(B) no. 18H01136.

Acknowledgements. The author would like to thank Dr Koya Sakakibara for helping me to make the contour plots of the streamfunction and the vorticity on the surface of torus. The author also thanks anonymous referees for fruitful comments.

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
