## [Reviewer comments · Proceedings. Mathematical, Physical, and Engineering Sciences]

Review History

RSPA-2018-0666.R0 (Original submission)

Review form: Referee 1

Is the manuscript an original and important contribution to its field?

Yes

Is the paper of sufficient general interest?

Yes

Is the overall quality of the paper suitable?

Yes

Quality of the paper

An excellent paper making an important contribution to the field: should be published.

Can the paper be shortened without overall detriment to the main message?

No

Do you think some of the material would be more appropriate as an electronic appendix?

No

For papers with colour figures – is colour essential?

Yes

If there is supplementary material, is this adequate and clear?

Not applicable

Are there details of how to obtain materials and data, including any restrictions that may apply?

Not applicable

Do you have any ethical concerns with this paper?

No

Recommendation?

Accept as is

Comments to the Author(s)

An excellent paper containing some excellent results.

Review form: Referee 2

Is the manuscript an original and important contribution to its field?

Yes

Is the paper of sufficient general interest?

Yes

Is the overall quality of the paper suitable?

Yes

Quality of the paper

A good paper worth publishing in Proceedings.

Can the paper be shortened without overall detriment to the main message?

No

Do you think some of the material would be more appropriate as an electronic appendix?

No

For papers with colour figures – is colour essential?

Yes

If there is supplementary material, is this adequate and clear?

Not applicable

Are there details of how to obtain materials and data, including any restrictions that may apply?

Yes

Do you have any ethical concerns with this paper?

No

Recommendation?

Accept with minor revision (please list in comments)

Comments to the Author(s)

This paper finds the analogous planar Stuart vortex distribution on the surface of a torus. In particular, the author considers the three-vortex solution as a simple realization of such fluid flows. Although the reviewer didn't check the calculations in detail, the result seems reasonable. Overall, he recommends accepting the paper with the following possible improvements.

1. The author considers probably the most simple one $f(z) = \frac{sn(z)}{cn(z)}$ which gives always the two same strength vortices plus one opposite sign vortex configuration as in the paper. More interesting and realistic will be the general n-vortex solution on the torus. This may be commented briefly in this paper and will be a future research topic.
2. The pictures in the paper are flat and do not show the physical real flows on the torus. It will be more helpful to the reader if some of the real physical flow patterns are presented. In particular, the dependence of α of the flow pattern is not clearly seen in current pictures.

Decision letter (RSPA-2018-0666.R0)

18-Dec-2018

Dear Professor Sakajo

The Editor of Proceedings A has now received comments from referees on the above paper and would like you to revise it in accordance with their suggestions which can be found below (not including confidential reports to the Editor).

Please submit a copy of your revised paper within four weeks - if we do not hear from you within this time then it will be assumed that the paper has been withdrawn. In exceptional circumstances, extensions may be possible if agreed with the Editorial Office in advance.

Please note that it is the editorial policy of Proceedings A to offer authors one round of revision in which to address changes requested by referees. If the revisions are not considered satisfactory by the Editor, then the paper will be rejected, and not considered further for publication by the journal. In the event that the author chooses not to address a referee's comments, and no scientific justification is included in their cover letter for this omission, it is at the discretion of the Editor whether to continue considering the manuscript.

- Ethics statement
- Data accessibility
- Competing interests
- Authors' contributions
- Acknowledgements
- Funding statement

See <http://royalsocietypublishing.org/instructions-authors#question3> for further details.

To revise your manuscript, log into <http://mc.manuscriptcentral.com/prsa> and enter your Author Centre, where you will find your manuscript title listed under "Manuscripts with Decisions." Under "Actions," click on "Create a Revision." Your manuscript number has been appended to denote a revision.

You will be unable to make your revisions on the originally submitted version of the manuscript. Instead, revise your manuscript and upload a new version through your Author Centre.

When submitting your revised manuscript, you will be able to respond to the comments made by the referee(s) and upload a file "Response to Referees" in "Section 6 - File Upload". Please use this to document how you have responded to the comments, and the adjustments you have made. In order to expedite the processing of the revised manuscript, please be as specific as possible in your response to the referee(s).

IMPORTANT: Your original files are available to you when you upload your revised manuscript. Please delete any unnecessary previous files before uploading your revised version.

When revising your paper please ensure that it remains under 28 pages long. In addition, any pages over 20 will be subject to a charge (£150 + VAT (where applicable) per page). Your paper has been ESTIMATED to be 17 pages.

Once again, thank you for submitting your manuscript to Proc. R. Soc. A and I look forward to receiving your revision. If you have any questions at all, please do not hesitate to get in touch.

Yours sincerely
Alice Power
Publishing Editor
Proceedings A
proceedingsa@royalsociety.org

Reviewer(s)' Comments to Author:

Referee: 1

Comments to the Author(s)
An excellent paper containing some excellent results.

Referee: 2

Comments to the Author(s)
This paper finds the analogous planar Stuart vortex distribution on the surface of a torus. In particular, the author considers the three-vortex solution as a simple realization of such fluid flows. Although the reviewer didn't check the calculations in detail, the result seems reasonable. Overall, he recommends accepting the paper with the following possible improvements.

1. The author considers probably the most simple one $f(z) = \frac{\sigma(z)}{\rho(z)}$ which gives always the two same strength vortices plus one opposite sign vortex configuration as in the

paper. More interesting and realistic will be the general n -vortex solution on the torus. This may be commented briefly in this paper and will be a future research topic.

2. The pictures in the paper are flat and do not show the physical real flows on the torus. It will be more helpful to the reader if some of the real physical flow patterns are presented. In particular, the dependence of α of the flow pattern is not clearly seen in current pictures.

Author's Response to Decision Letter for (RSPA-2018-0666.R0)

See Appendices A & B.

RSPA-2018-0666.R1 (Revision)

Review form: Referee 1

Is the manuscript an original and important contribution to its field?

Yes

Is the paper of sufficient general interest?

Yes

Is the overall quality of the paper suitable?

Yes

Quality of the paper

An excellent paper making an important contribution to the field: should be published.

Can the paper be shortened without overall detriment to the main message?

No

Do you think some of the material would be more appropriate as an electronic appendix?

No

For papers with colour figures - is colour essential?

Yes

If there is supplementary material, is this adequate and clear?

Not applicable

Are there details of how to obtain materials and data, including any restrictions that may apply?

Not applicable

Do you have any ethical concerns with this paper?

No

Recommendation?

Accept as is

Comments to the Author(s)

As per my earlier review, this is an excellent paper. It should now be published as is.

Review form: Referee 2

Is the manuscript an original and important contribution to its field?

Yes

Is the paper of sufficient general interest?

Yes

Is the overall quality of the paper suitable?

Yes

Quality of the paper

An excellent paper making an important contribution to the field: should be published.

Can the paper be shortened without overall detriment to the main message?

No

Do you think some of the material would be more appropriate as an electronic appendix?

No

For papers with colour figures - is colour essential?

No

If there is supplementary material, is this adequate and clear?

Not applicable

Are there details of how to obtain materials and data, including any restrictions that may apply?

Yes

Do you have any ethical concerns with this paper?

No

Recommendation?

Accept as is

Comments to the Author(s)

The paper is revised as suggested by the reviewer.

Decision letter (RSPA-2018-0666.R1)

Dear Professor Sakajo

On behalf of the Editor, I am pleased to inform you that your manuscript entitled "Exact solution to a Liouville equation with Stuart vortex distribution on the surface of a torus" has been accepted in its final form for publication in Proceedings A.

Our Production Office will be in contact with you in due course. You can expect to receive a proof of your article soon. Please contact the office to let us know if you are likely to be away from e-mail in the near future. If you do not notify us and comments are not received within 5 days of sending the proof, we may publish the paper as it stands.

Open access

You are invited to opt for open access, our author pays publishing model. Payment of open access fees will enable your article to be made freely available via the Royal Society website as soon as it is ready for publication. For more information about open access please visit http://royalsocietypublishing.org/site/authors/open_access.xhtml. The open access fee for this journal is £1700/\$2380/€2040 per article. VAT will be charged where applicable.

Note that if you have opted for open access then payment will be required before the article is published – payment instructions will follow shortly. If you wish to opt for open access then please inform the editorial office (proceedingsa@royalsociety.org) as soon as possible.

Your article has been estimated as being 16 pages long. Our Production Office will inform you of the exact length at the proof stage.

Proceedings A levies charges for articles which exceed 20 printed pages. (based upon approximately 540 words or 2 figures per page). Articles exceeding this limit will incur page charges of £150 per page or part page, plus VAT (where applicable).

Under the terms of our licence to publish you may post the author generated postprint (ie. your accepted version not the final typeset version) of your manuscript at any time and this can be made freely available. Postprints can be deposited on a personal or institutional website, or a recognised server/repository. Please note however, that the reporting of postprints is subject to a media embargo, and that the status the manuscript should be made clear. Upon publication of the definitive version on the publisher's site, full details and a link should be added.

You can cite the article in advance of publication using its DOI. The DOI will take the form: 10.1098/rspa.XXXX.YYYY, where XXXX and YYYY are the last 8 digits of your manuscript number (eg. if your manuscript number is RSPA-2017-1234 the DOI would be 10.1098/rspa.2017.1234).

For tips on promoting your accepted paper see our blog post: <https://blogs.royalsociety.org/publishing/promoting-your-latest-paper-and-tracking-your-results/>

Thank you for your submission. On behalf of the Editors of the journal, we look forward to your continued contributions to the Journal.

Best wishes
Alice Power
Publishing Editor

Proceedings A Editorial Office
proceedingsa@royalsociety.org

on behalf of
Dr Nader Masmoudi
Board Member
Proceedings A

Reviewer(s)' Comments to Author:

Referee: 1

Comments to the Author(s)

As per my earlier review, this is an excellent paper. It should now be published as is.

Referee: 2

Comments to the Author(s)

The paper is revised as suggested by the reviewer.

Appendix A

List of changes

We have revised the paper, “Exact solution to a Liouville equation with Stuart vortex distribution on the surface of a torus”, following the comments and suggestions by the second referee. The following is the list of changes. The revised parts are highlighted in red color.

- Following the second referee’s suggestion, I have commented on the future works to be investigated more clearly in the last paragraph of Section 5.
- I have added Figure 4 to the revised manuscript, showing contour plots of the streamfunction and the vorticity on the surface of torus with various aspect ratios. This will help the readers understand the real physical flow patterns more clearly.

Appendix B

Response to the second referee

I thank the referee for reviewing the paper, “Exact solution to a Liouville equation with Stuart vortex distribution on the surface of a torus”. I have revised the paper, following the suggestions by the referee. The changes made in the revised manuscript are highlighted in red color.

- *The author considers probably the most simple one $f(z) = \frac{sn(z)}{cn(z)}$ which gives always the two same strength vortices plus one opposite sign vortex configuration as in the paper. More interesting and realistic will be the general n -vortex solution on the torus. This may be commented briefly in this paper and will be a future research topic.*

⇒ I totally agree with the referee’s suggestion. This is commented in the last paragraph of Section 5 clearly.

- *The pictures in the paper are flat and do not show the physical real flows on the torus. It will be more helpful to the reader if some of the real physical flow patterns are presented. In particular, the dependence of α of the flow pattern is not clearly seen in current pictures.*

⇒ I have added Figure 4, showing contour plots of the streamfunction and the vorticity on the surface of torus with various aspect ratios. The figure will help the readers understand the real physical flow patterns, in which the dependence of the aspect ratio α is clearly visualised.